# UALM : Unified Audio Language Model for Understanding, Generation and Reasoning

**Jinchuan Tian**[1†*], **Sang-gil Lee**[2*], **Zhifeng Kong**[2*], **Sreyan Ghosh**[23†], **Arushi Goel**[2],
**Chao-Han Huck Yang**[2], **Wenliang Dai**[2], **Zihan Liu**[2], **Hanrong Ye**[2], **Shinji Watanabe**[1],
**Mohammad Shoeybi**[2], **Bryan Catanzaro**[2], **Rafael Valle**[2], **Wei Ping**[2]
jinchuat@andrew.cmu.edu, {sanggill,zkong,wping}@nvidia.com

## ABSTRACT

Recent advances in the audio language modeling (ALM) domain tackle audio understanding and text-to-audio generation as separate tasks. Very few studies attempt to unify these tasks – an essential step toward advanced multimodal reasoning. This paper introduces **U**nified **A**udio **L**anguage **M**odel (UALM), which aims to unify audio understanding, text-to-audio generation, and multimodal reasoning in a single model. To achieve this goal, we first present UALM-Gen, a text-to-audio language model that directly predicts audio tokens and is comparable to state-of-the-art diffusion-based models. We then demonstrate, using proper data blending, training recipes, and inference techniques, that our single UALM model matches the quality of state-of-the-art specialized models in audio understanding, text-to-audio generation, and text reasoning. Furthermore, we present UALM-Reason, a multimodal reasoning model that utilizes both text and audio in the intermediate thinking steps to facilitate complex generation tasks. To our knowledge, this is the first demonstration in audio research of cross-modal generative reasoning, with its effectiveness confirmed by subjective evaluations. [‡]

## 1 INTRODUCTION

Human auditory intelligence is characterized by two fundamental capabilities: perception (understanding) and production (generation). This duality is not merely conceptual; neuro-scientific evidence reveals a profound connections between these functions, where impairment in one often corresponds to a deficit in the other (Liberman et al., 1967; Hickok & Poeppel, 2007; Rizzolatti & Craighero, 2004). Furthermore, resolving complex acoustic challenges requires a sophisticated reasoning process that is inherently multimodal (McGurk & MacDonald, 1976; Leman, 2007; Denes & Pinson, 1993; Liberman & Mattingly, 1985). This cognitive loop, which often transcends purely textual representation, is exemplified by a music composer who iteratively creates a piece (generation), critically listens to it (understanding), and refines it (self-reflection) (Hallam et al., 2016). This human paradigm of tightly inte-

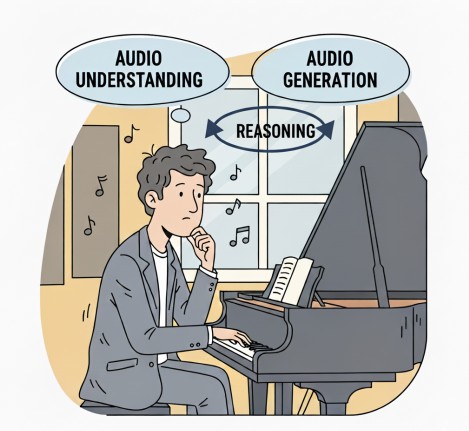

Figure 1: Humans need understanding, generation, and reasoning to handle complex tasks, like composing music.

grating understanding, generation, and reasoning motivates our work and suggests that unifying these three pillars is a crucial step toward advanced and general audio intelligence.

However, realizing the vision of unified audio intelligence faces significant challenges rooted in the prevailing research landscape. **First**, prior works predominantly tackled audio understanding

---

[*]Equal Contribution. CMU[1]. NVIDIA[2]. UMD[3]. †: Work done during an internship at NVIDIA.

[‡]Demo samples: https://research.nvidia.com/labs/adlr/UALM. Code: https://github.com/NVIDIA/audio-intelligence/tree/main/UALM.

and text-to-audio generation as separate tasks. This gap is further entrenched by a divergence in modeling paradigms: understanding tasks are mainly addressed with auto-regressive large language models (Goel et al., 2025; Xu et al., 2025), while state-of-the-art generation models are mostly based on diffusion models (Lee et al., 2024; Fei et al., 2024; Valle et al., 2025). **Second**, reasoning within the audio domain remains highly under-explored. Current reasoning studies are confined to the text-only reasoning trajectory in service of audio understanding tasks (Xie et al., 2025; Diao et al., 2025; Goel et al., 2025; Rouditchenko et al., 2025). However, it is as important to have the ability to reason for guided generation or problem solving that requires a multimodal thinking process. In this work, we aim to close this gap and present the **U**nified **A**udio **L**anguage **M**odel (UALM) for supporting audio understanding, text-to-audio generation, and multimodal reasoning concurrently in a unified manner: UALM handles all tasks via a single language model and is capable of reasoning through an interleaved and flexible understanding-generation chain-of-thought, a mechanism central to human creativity.

**The first challenge is to achieve high-quality text-to-audio via a language model**. Prior diffusion-based text-to-audio models (Evans et al., 2024; Lee et al., 2024; Fei et al., 2024) outperform autoregressive models (Yang et al., 2023; Copet et al., 2024) in quality, potentially due to better inductive bias (Vastola, 2025) and data efficiency (Prabhudesai et al., 2025). We discover several important findings for successful text-to-audio generation via a decoder-only language model: $(i)$ in terms of data scaling, we need an order of magnitude more audio data than diffusion-based models; $(ii)$ we show that training and sampling with classifier-free guidance (Ho & Salimans, 2022) – a technique widely used in diffusion models but rarely in multimodal language models – is critical to high-quality generation in our model; $(iii)$ we find a versatile codec (Ye et al., 2025) suitable for audio token generation, and it is crucial to train and infer audio tokens with the delay pattern (Copet et al., 2024) for efficiency; $(iv)$ we find that applying a self-adaptation stage followed by direct preference optimization (DPO) (Rafailov et al., 2023) can further improve our audio generation quality and aesthetics. With these techniques, we introduce UALM-Gen for text-to-audio generation, and experiments confirm that UALM-Gen achieves quality comparable to frontier diffusion-based models.

**The second challenge is to unify all three tasks in a single language model.** While there are several unified generation and understanding models in the vision or speech domains (Wu et al., 2024; Team, 2024; Tian et al., 2025a), in our preliminary experiments we find their recipes do not directly apply to the broader audio domain, and it is hard to balance between different tasks. To tackle these challenges, we $(i)$ carefully design the data blending ratios and up-weight generation data due to its slower convergence, and $(ii)$ apply a modality alignment stage to warm up the MLP adapter layers and all token embeddings before unfreezing the full language model backbone. With these, we present a single UALM that is comparable to state-of-the-art specialized models in each of the three domains: text problem solving, audio understanding, and text-to-audio generation.

**The third challenge is to achieve generative multimodal reasoning beyond the text domain.** The formal definition, training data, and training recipes for *reasoning in audio generation* are not well-defined yet. This work takes the first attempt toward this challenging task by investigating three specific steps. $(i)$ We first introduce *rich captions* – structured and comprehensive descriptions of audio – as an intermediate blueprint for generation. $(ii)$ We then enable the model to chat with the user to consolidate all details for generation. $(iii)$ We further guide the model to understand and critique its self-generated content, and produce an improved follow-up generation. To achieve these abilities, we present a principled data curation and training recipe, and introduce UALM-Reason. Experiments show reasoning in generation improves controllability towards nuanced prompts. To our knowledge, UALM-Reason is one of the earliest works to achieve audio reasoning with the multimodal thinking trajectory beyond the text-only domain.

In summary, this work presents the following contributions toward general audio intelligence. $(i)$ We present UALM-Gen, an LLM that predicts audio tokens and achieves state-of-the-art text-to-audio generation quality (§2.2). $(ii)$ We introduce UALM, a single LLM that unifies audio understanding, text-to-audio generation, and text-only tasks. UALM achieves competitive results across all three domains (§2.3) $(iii)$ We demonstrate UALM-Reason, a reasoning model focused on multimodal reasoning beyond the text domain. UALM-Reason unifies reasoning across understanding and generation tasks and demonstrates better controllability (§2.4). $(iv)$ We present practically effective data strategies, training recipes, and inference techniques to enable unified audio language modeling with ablations.

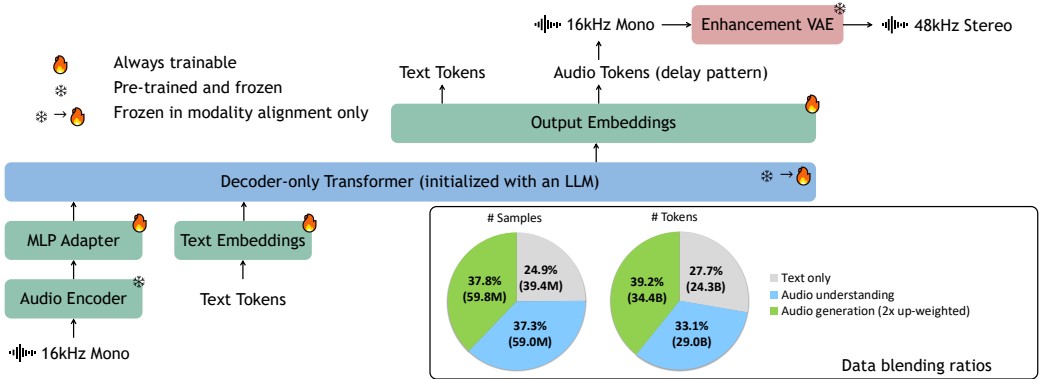

Figure 2: UALM architecture overview and the multimodal pre-training data blending ratios.

# 2 UNIFIED AUDIO LANGUAGE MODEL (UALM)

## 2.1 ARCHITECTURE

The architecture of UALM is presented in Fig.2, which extends a pre-trained decoder-only text LLM with audio inputs and outputs.

**Audio Input and Output:** We adopt the well-established *Encoder-Adapter-LLM* architecture (Liu et al., 2023b; Goel et al., 2025) for audio input, which connects the modules with continuous representations and avoids the input information loss caused by discrete tokenization. We adopt the acoustic encoder from Goel et al. (2025) for audio input, which operates at 25Hz frame rate and a sliding window chunk size of 30 seconds. The adapter is a single-layer MLP.

Audio output is achieved by predicting discrete audio codec tokens, a common practice adopted by previous works (Tian et al., 2025a; Copet et al., 2024; Yuan et al., 2025). We use X-codec (Ye et al., 2025) which operates at 50Hz frame rate. Each frame is discretized via residual vector quantization (RVQ) (Zeghidour et al., 2021) producing 8 tokens per frame. We employ the *delay* pattern (Copet et al., 2024) for intra-frame auto-regression of RVQ, a technique proven effective in prior audio generation research (Yang et al., 2023). See Appendix B.1 for details of RVQ and the delay pattern.

Both the acoustic encoder's input and the audio codec decoder's output operate at monophonic 16kHz waveform. We additionally introduce an enhancement VAE module to improve the output waveform to 48kHz stereo with better perceptual quality (Appendix B.2).

**Initialization:** UALM is initialized from Qwen2.5-7B (Yang et al., 2024), a text LLM with reasoning capabilities. We extend its vocabulary to accommodate additional audio tokens from the audio codec. The additional audio embeddings are randomly initialized together with the MLP adapter.

**Implementation Details:** In all training stages, we only compute the loss over the model output tokens, which could be either in text or audio domains. We consider that one audio frame is equivalently important to one text token[1]. Sequence packing (Krell et al., 2021) is crucial during pre-training to accommodate interleaved samples of varying lengths and target modalities, thereby avoiding skewed sample and/or length distributions within a mini-batch and stabilizing training.

## 2.2 UALM-GEN: LANGUAGE MODEL-BASED AUDIO GENERATION

Using an auto-regressive LM for audio understanding (Goel et al., 2025) and text reasoning (Liu et al., 2024d; Bercovich et al., 2025b) has been well explored in prior works. However, LM-based text-to-audio generation, although proven feasible (Yang et al., 2023; Copet et al., 2024), is empirically found to be inferior to the diffusion-based counterparts (Evans et al., 2024; Lee et al., 2024). This section builds UALM-Gen, which demonstrates that the LM paradigm is simple, scalable, and achieves frontier results for audio generation. We build UALM-Gen with 1.5B parameters, with its weights initialized from Qwen2.5-1.5B.

---

[1]Since each audio frame contains 8 tokens, we scale the loss of each audio token by 1/8.

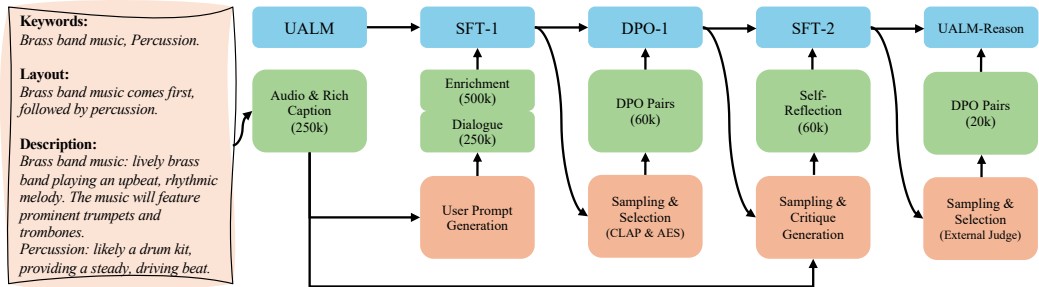

Figure 3: Rich caption example (left) and post-training workflow (right).

**Removal of External Text Encoder for Caption Embedding**: For both LM and diffusion approaches, A prevalent practice in audio generation (Kreuk et al., 2022; Copet et al., 2024; Lee et al., 2024; Hung et al., 2024) is to cross-attend the caption embedding from an external text encoder (e.g., T5 (Xue et al., 2022)), which is not compatible with our architecture in Section 2.1. For the first time, we show that LM-based audio generation can process text prompts as standard BPE tokens by initializing from a pre-trained text LLM.

**Data Scaling**: Our investigation reveals that LM-based audio generation needs significantly more data than diffusion-based methods. While previous diffusion models have achieved strong results on relatively small data volumes (often <2M samples or <4k hours (Lee et al., 2024)), we found that the LM-based approach cannot reach competitive quality at a similar data scale. We therefore scaled our training data volume up to 30M samples (approx. 80k hours and 17B tokens), a crucial step that enabled UALM-Gen to match the result of frontier diffusion-based counterparts.

**Classifier-Free Guidance (CFG)**: CFG (Ho & Salimans, 2022) is a widely used inference-time technique in generative models that enhances instruction following, which generates the sequence $y_{1:T}$ based on an interpolation between the conditional and unconditional distribution:

$$\pi_\theta^{\text{CFG}}(y_t|y_{1:t-1}, x) = \lambda \cdot \pi_\theta(y_t|y_{1:t-1}, x) + (1-\lambda) \cdot \pi_\theta(y_t|y_{1:t-1}, \emptyset),   \quad (1)$$

where $\lambda \geq 1$ is the CFG hyper-parameter and $\emptyset$ means null condition. Consistent with findings from previous LM (Kreuk et al., 2022; Copet et al., 2024; Hussain et al., 2025) and diffusion models (Lee et al., 2024; Hung et al., 2024), we show that CFG is also an important component for improving the quality of LM-based audio generation.

**Direct Preference Optimization (DPO)**: After the base model is trained with cross-entropy loss, we further conduct DPO training. DPO (Rafailov et al., 2023) is an offline reinforcement learning algorithm, first applied in text models (Rafailov et al., 2023) and later to audio generation in diffusion (Hung et al., 2024) and LMs (Hussain et al., 2025; Tian et al., 2025b). It optimizes a model using a preference pair of winning and losing samples $(y_w, y_l)$ for a given prompt $x$:

$$\mathcal{L}_{\text{DPO}}(\pi_\theta) = -\mathbb{E}_{(x,y_w,y_l)\sim\mathcal{D}}\left[\log\sigma\left(\beta\log\frac{\pi_\theta(y_w|x)}{\pi_{\text{ref}}(y_w|x)} - \beta\log\frac{\pi_\theta(y_l|x)}{\pi_{\text{ref}}(y_l|x)}\right)\right],  \quad (2)$$

where $\pi_\theta$ is the trained model; $\pi_{\text{ref}}$ is the reference model initialized from $\pi_\theta$ and frozen. $\sigma$ is *sigmoid* function and $\beta$ is a hyper-parameter. To obtain such preference pairs, we generate 10 synthetic samples for each prompt and select the winning and losing samples with a judge model. We found it necessary to first adapt the model to the self-generated winning samples using cross-entropy loss. Since the base model was pre-trained exclusively on natural audio, directly applying DPO to synthetic pairs can induce out-of-domain issue and subsequently performance degradation—issues that, in our experiments, could not be mitigated solely by tuning the hyperparameter $\beta$. It is also found that the cross-entropy regularizer helps to reduce the divergence (i.e., $\pi_\theta(y_w|x) - \pi_{\text{ref}}(y_w|x)$) from the base model. Experimental results are in §3.2.

## 2.3 UALM: UNIFIED AUDIO UNDERSTANDING AND GENERATION PRE-TRAINING

Having established a strong foundation for single-task audio generation, we proceed to the continued multimodal pre-training of UALM from the text LLM. This phase is to simultaneously cultivate capabilities in audio understanding, audio generation, and text-based reasoning within a single model.

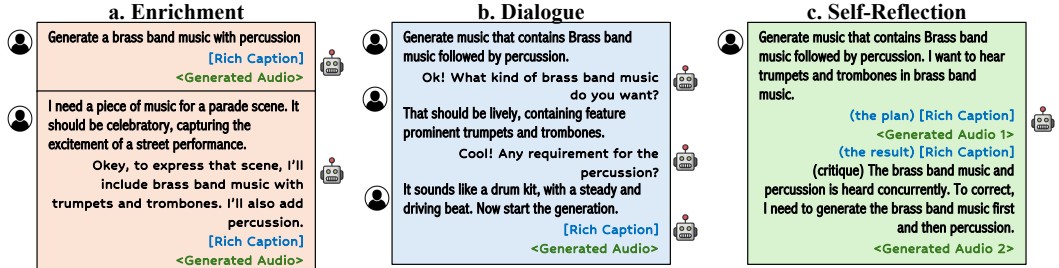

Figure 4: Demos: audio generation reasoning and joint understanding-generation reasoning.

**Pre-training Data Mixture**: To achieve this unification, we create a comprehensive data mixture that fuses datasets from all three target domains. Audio understanding and generation data are combined with text-only reasoning data, enabling the model to develop a shared representational space. The blending ratios of our data mixture are in Fig.2.

**Modality Alignment Stage**: While our single-task generation model could be trained effectively without a specific curriculum, we found that a dedicated modality alignment stage is critical for the success of our unified pre-training, an observation consistent with prior work (Wu et al., 2025b). In this initial phase, we freeze the Transformer body and acoustic encoder, updating only the MLP adapter and audio embedding tables using a small number of steps but a large batch size. After this stage, we unfreeze all parameters for all follow-up training stages, except the acoustic encoder.

## 2.4 UALM-REASON: POST-TRAINING FOR MULTIMODAL REASONING

To unlock more advanced cognitive abilities, we advance the pre-trained UALM to its reasoning-enhanced version, UALM-Reason, through a dedicated post-training procedure. This stage introduces a multimodal Chain-of-Thought (CoT) paradigm where the model generates intermediate multimodal reasoning steps, in audio and/or text, to deconstruct a user's request, formulate a detailed generation plan, and even critique its own output before producing the final audio.

We note that the reasoning capabilities for audio understanding have already been established during pre-training[2]. This text-based reasoning has already been addressed in prior works (Wu et al., 2025c) and is not a contribution of this work. Thus, our primary focus in post-training shifts to pioneer reasoning for audio *generation*. This is achieved through a two-stage interleaved SFT-DPO recipe, which instills three novel reasoning patterns: enrichment, dialogue, and self-reflection.

### 2.4.1 MULTIMODAL REASONING FOR AUDIO GENERATION

Central to our approach is the concept of a *rich caption*—a structured and highly detailed textual description that serves as an intermediate blueprint for audio generation. Unlike conventional short prompts, a rich caption provides a comprehensive plan by specifying: *Keywords:* A list of the core acoustic events. *Layout:* The temporal arrangement of these events. *Description:* A detailed characterization of each event's acoustic properties. An example of the rich caption is in Fig.3.

This detailed intermediate representation provides nuanced guidance that is critical for high-fidelity and controllable audio synthesis. The following reasoning capabilities are designed to bridge the gap from simple and diverse user queries to this rich, machine-usable format.

**Enrichment:** User prompts are often abstract, short, and underspecified. The enrichment capability allows UALM-Reason to autonomously translate a user prompt into a detailed rich caption. The model faithfully incorporates all user-provided details while inferring and adding necessary specifics (e.g., environmental context, instrument textures) to create a complete acoustic scene. We further support the abstract user prompt that describes a scenario, a feeling, or a genre, rather than the exact audio events. For these imaginary prompts, the model would enrich with all suitable audio events and the corresponding details automatically. Examples are in Fig.4.a.

---

[2]The AF3 (Goel et al., 2025) data mixture for audio understanding contains massive reasoning samples for audio understanding. To preserve this capability during post-training, we uniformly sample the AF3 mixture to account for 20% volume during each SFT stage.

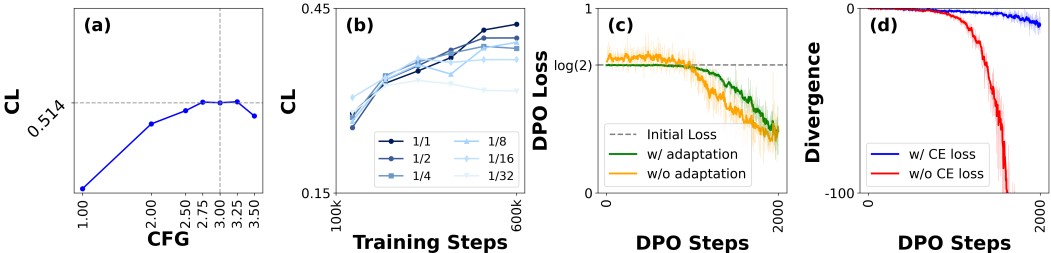

Figure 5: Statistics of UALM-Gen model. (a) The CLAP scores (CL) with various CFG $\lambda$; (b) The CLAP scores (CL) with various training data volume down-weighting; (c) the DPO loss w/o adaptation on synthetic data before DPO training; (d) the divergence $\pi_\theta(y_w|x) - \pi_{\text{ref}}(y_w|x)$ from the reference model w/o CE loss added in DPO training.

**Dialogue:** As an interactive alternative to enrichment, the model can engage in a multi-turn dialogue to collaboratively construct the rich caption. It actively queries the user for specific details, guiding them to provide the information needed for a successful generation, thereby resolving ambiguity before synthesis begins. Examples are in Fig.4.b.

**Self-Reflection:** This represents the most advanced form of reasoning, creating reasoning traces that leverage all audio understanding, generation, and text capabilities. The process unfolds as follows: *Generate:* Following *enrichment* or *dialogue* paradigm, the model first generates a rich caption and subsequently an audio clip based on a user prompt. *Understand:* It then "listens" to its own output and generates another *new* rich caption describing what it actually produced. *Critique & Refine:* The model compares the two rich captions (the plan vs. the result), identifies discrepancies or flaws in a textual *critique*, and uses this feedback to generate a second, improved audio clip. This *generate-understand-critique-refine* cycle, also known as self-reflection (Liu et al., 2024a), mimics human creative iteration and marks a significant step towards higher-level intelligence in multimodal models. Examples are in Fig.4.c.

### 2.4.2 Two-Stage SFT-DPO Training Recipe

UALM-Reason's reasoning capabilities are instilled via two sequential rounds of interleaved SFT-DPO curriculum.

**Round 1: Building Foundational Generation Reasoning.** This round focuses on teaching the *enrichment* and *dialogue* capabilities. We begin with 250k internal rich caption-audio pairs. For SFT, a text LLM is used to synthetically generate diverse user prompts and conversational dialogues that correspond to these rich captions, resulting in a 750k-sample training set for the `SFT-1` model. We then apply DPO to a 250k subset of these samples to create the `DPO-1` model: we only use the *keywords* from the rich caption to compute CLAP scores for preference ranking due to context length limitations of existing CLAP models. We obtain around 60k DPO pairs after threshold-based filtering, as in §3.2.

**Round 2: Enabling Self-Reflection.** The second round introduces the self-reflection capability. We create a new dataset from 60k samples from the first SFT round. For each sample, we use the `DPO-1` model to generate the initial audio, then curate a rich caption for it. A text LLM then generates a textual *critique* comparing the planned and actual rich captions, highlighting the most salient flaw and suggesting a fix. This self-reflection data is combined with the first-round SFT data to train the `SFT-2` model. Finally, we perform a targeted DPO step on 20k samples (with and without self-reflection). For preference selection, we choose the sample that better adheres to the detailed instructions in the original rich caption. This final optimization step yields the **UALM-Reason** model.

## 3 Experiments

### 3.1 Experimental Setup

**Data:** Starting from a text LLM, our continued pre-training data is a mixture designed to support audio understanding, generation, and text-only reasoning. For all audio understanding tasks, our

Table 1: Audio Generation results of UALM-Gen (§2.2) and UALM (§2.3) compared to LM-based and diffusion-based baselines. 5-scale subjective scores (OVL, REL) 95% CI ≈ 0.10. **Bold** indicates best, underline second-best.

| Model | SongDescriber | | | | | | | AudioCaps | | | | | | |
|---|---|---|---|---|---|---|---|---|---|---|---|---|---|---|
| | FD↓ | KL↓ | IS↑ | CL↑ | AES↑ | OVL↑ | REL↑ | FD↓ | KL↓ | IS↑ | CL↑ | AES↑ | OVL↑ | REL↑ |
| Ground Truth | 0 | 0 | 1.88 | 0.48 | 7.20 | 4.10 | 4.03 | 0 | 0 | 13.49 | 0.62 | 4.50 | 3.91 | 3.96 |
| MusicGen-stereo-L (Copet et al., 2024) | 228.94 | 0.84 | 1.69 | 0.36 | 6.64 | 3.91 | 3.97 | — | — | — | — | — | — | — |
| AudioGen-M (Kreuk et al., 2022) | — | — | — | — | — | — | — | 149.70 | 1.35 | 12.26 | 0.57 | 4.21 | 3.83 | 3.89 |
| MAGNeT-M (Ziv et al., 2024) | 191.49 | 0.70 | 1.47 | 0.41 | 6.65 | 3.84 | 3.86 | 149.68 | 1.86 | 7.73 | 0.46 | 4.08 | 3.68 | 3.72 |
| AudioLDM2-L (Liu et al., 2024b) | 331.73 | 0.68 | 1.96 | 0.45 | 6.31 | 3.83 | 3.80 | 121.63 | 1.72 | 8.59 | 0.52 | 4.31 | 3.85 | 3.75 |
| TangoFlux (Hung et al., 2024) | 235.61 | 0.71 | 1.70 | 0.41 | 6.46 | 3.80 | 3.89 | 103.04 | **1.02** | 15.13 | 0.65 | 4.42 | 3.72 | 3.93 |
| Stable Audio Open (Evans et al., 2024) | 138.58 | 1.01 | **2.25** | 0.42 | 6.37 | 3.92 | 3.97 | 100.93 | 2.22 | 11.80 | 0.35 | 4.47 | 3.81 | 3.80 |
| ETTA (Lee et al., 2024) | 95.66 | 0.80 | 2.15 | 0.44 | 6.71 | 3.92 | 3.93 | 80.13 | 1.22 | 14.36 | 0.54 | 4.51 | 3.73 | **3.94** |
| UALM-Gen (Ours) | **74.43** | 0.63 | 1.87 | **0.54** | **7.36** | **4.07** | 3.96 | 75.14 | 1.19 | 14.52 | **0.65** | **5.08** | 3.79 | 3.92 |
| UALM (Ours) | 83.69 | **0.59** | 2.00 | **0.54** | 7.28 | 3.97 | **3.99** | **65.87** | 1.35 | **15.62** | 0.62 | 4.92 | **3.89** | 3.86 |

dataset is identical to that used by AF3 (Goel et al., 2025), which already contains reasoning content. We curate a large-scale audio generation dataset of 30M text-audio pairs with a length of 10s each. Notably, the majority of the text captions are pseudo-labels generated by open-source audio captioning models (Xu et al., 2025; Ghosh et al., 2025; Goel et al., 2025). The convergence of audio generation is slow (§2.3) and its data is comparatively small in volume, so we empirically up-sample it by 2x. To preserve and enhance the model's native text reasoning capabilities, we integrated 21 million samples of math and code reasoning data from Liu et al. (2024d) and Bercovich et al. (2025b). An additional 3 million in-house text samples were included to bolster commonsense knowledge. Other data usage in post-training is described in §3.2 and §2.4.

**Optimization and Inference:** The model was pre-trained on a compute cluster of 16 nodes, each equipped with 8 NVIDIA A100 80GB GPUs. We utilized a per-GPU batch size of 5,000 tokens and trained the model for a total of 660,000 steps. We always use greedy search for text generation. For audio generation, we use top-k sampling. Our detailed configurations are in Appendix C.1.

**Objective Evaluation**: For audio generation, we evaluate our model on the AudioCaps (Kim et al., 2019) and SongDescriber (Manco et al., 2023) test sets. We follow standard evaluation protocols (Evans et al., 2024; Lee et al., 2024) : (1) Frechet distance (FD) using OpenL3 (Cramer et al., 2019) at 44.1kHz; (2) Kullback–Leibler divergence (KL) using PaSST (Koutini et al., 2022) at 32kHz; (3) Inception Score (IS) using PANNs (Kong et al., 2020) at 16kHz; (4) CLAP scores (CL) using LAION-CLAP (Wu et al., 2023) at 48kHz; (5) AudioBox-Aesthetic score (AES) (Tjandra et al., 2025) using an average of (CE, CU, PC, PQ) at 16kHz. For audio understanding, we perform evaluations on MMAU(Sakshi et al., 2024) and MMAR (Ma et al., 2025). For text-only evaluations, we always use the "reasoning mode" and test its zero-shot accuracy on MMLU (Hendrycks et al., 2020), GSM8K (Cobbe et al., 2021), and HumanEval (Chen et al., 2021), which corresponds to common sense, math, and code capability, respectively.

**Subjective Evaluation**: For audio generation, we additionally conduct subjective human evaluations of 5-scale mean opinion scores using mechanical turk, following established practices (Kreuk et al., 2022; Copet et al., 2024; Liu et al., 2023a; Lee et al., 2024): (1) OVL: an overall quality of sample without seeing captions; (2) REL: a relevance of the sample to the provided caption.

## 3.2 LANGUAGE MODEL-BASED TEXT-TO-AUDIO GENERATION RESULTS

With the base UALM-Gen model trained with cross-entropy loss, we first show that CFG is necessary for model inference. As suggested in Fig.5.a, audio generation without CFG encounters severe degradation. As defined in §2.2, we find the weight $\lambda = 3.0$ for CFG is optimal. For top-k sampling, we constantly use $k = 20$ without temperature rescaling.

Secondly, in Fig.5.b, we show the impact of data scaling where we reduce the data volume down to 1/32 of its full size (30M). The result indicates that data scaling is necessary for the success of the LM-based approach. Note that overfitting is clearly observed with 1/32 data down-sampling, where the data volume is comparable to prior state-of-the-art diffusion model ETTA (Lee et al., 2024) (1.0M vs. 1.3M). Our finding aligns with Prabhudesai et al. (2025), comparing scaling laws for auto-regressive models trained with cross-entropy and diffusion-based models.

Table 2: Audio understanding results of UALM (§2.3) versus open-sourced understanding models.

| Model | Base Model | MMAU-v05.15.25 | | | | MMAR |
| --- | --- | --- | --- | --- | --- | --- |
| | | Sound↑ | Music↑ | Speech↑ | Mean↑ | Mean↑ |
| GAMA-IT (Ghosh et al., 2024) | LLaMA-2 (7B) | 32.7 | 22.4 | 11.6 | 22.2 | 17.4 |
| SALMONN (Tang et al., 2024) | Vicuna (13B) | 42.1 | 37.8 | 28.8 | 36.2 | 33.2 |
| Qwen2-Audio-Instruct (Chu et al., 2024) | Qwen2-Audio (7B) | 61.2 | 55.7 | 55.4 | 57.4 | 30.0 |
| DeSTA2.5-Audio (Lu et al., 2025) | Llama3.1-8B-Instruct | 66.8 | 57.1 | 71.9 | 65.2 | — |
| Audio Reasoner (Xie et al., 2025) | Qwen2-Audio Instruct (7B) | 67.3 | 61.5 | 62.5 | 63.8 | 36.8 |
| Step-Audio-2 (Wu et al., 2025a) | Unknown (<130B) | 80.6 | 68.2 | 72.8 | 73.9 | — |
| Qwen2.5-Omni (Xu et al., 2025) | Qwen2.5 (7B) | 76.8 | 67.3 | 68.9 | 71.0 | 56.7 |
| Audio Flamingo 3 (Goel et al., 2025) | Qwen2.5 (7B) | 76.7 | 73.3 | 64.9 | 72.3 | 58.5 |
| UALM (Ours) | Qwen2.5 (7B) | 77.9 | 77.6 | 66.7 | 74.1 | 55.2 |

Table 3: Text capability of prior unified multimodal language models (in the vision domain) and our UALM. Our model is initialized from *Qwen2.5-7B*.

| Model | MMLU↑ | GSM8K↑ | HumanEval↑ | Mean↑ |
| --- | --- | --- | --- | --- |
| OpusLM (Tian et al., 2025a) | 52.5 | - | - | - |
| Liquid-7B (Wu et al., 2024) | 56.0 | - | - | - |
| Chameleon-7B (Team, 2024) | 52.1 | - | - | - |
| Qwen2.5-7B-Instuct (Yang et al., 2024) | 74.5 | 91.6 | 84.8 | 83.6 |
| UALM (Ours) | 71.6 | 92.1 | 81.1 | 81.6 |

Thirdly, we conduct DPO on the UALM-Gen base model. We sample 250k prompts uniformly from the pre-training data and generate 10 audio clips for each prompt. We then select the preference pairs using CL and (CE, CU, PC, PQ) metrics[3], which ultimately yields 50k pairs. As the base UALM-Gen is trained on real audio, it is found necessary to first adapt it to synthetic audio by fine-tuning on the winning examples (typically 1k steps). Without adaptation, the DPO loss would spike in the early training phase before convergence as in Fig.5.c. DPO training causes divergence from the base model, which could be alleviated by enforcing the cross-entropy loss over the winning samples, together with the DPO loss, as in Fig.5.d.

Ultimately, Tab. 1 shows that UALM-Gen outperforms previous LM-based approaches (Kreuk et al., 2022; Copet et al., 2024) and achieves competitive results to leading diffusion models, including TangoFlux (Hung et al., 2024), Stable Audio Open (Evans et al., 2024), and ETTA (Lee et al., 2024). Techniques introduced in §2.2 are further ablated in Tab.8.

### 3.3 MULTIMODAL PRE-TRAINING RESULTS

We demonstrate that the pre-trained base model UALM matches the quality of frontier special-ists in text-to-audio generation, audio under-standing, and text-based reasoning. The Au-dio generation capability is reported in Tab.1. Like UALM-Gen, the UALM outperforms or matches the quality of prior diffusion models. The audio understanding capability is reported in Tab.2. The results show that the accuracy of our model can match prior state-of-the-art open-source models like Audio Flamingo 3 and Qwen2.5-Omni. Finally, as presented in Tab 3, our UALM only encounters marginal degrada-

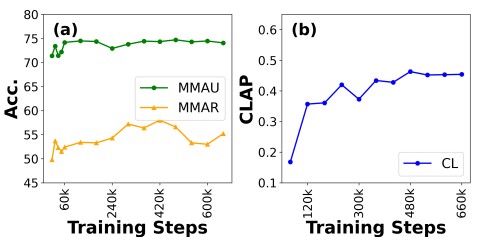

Figure 6: Audio understanding (a) and audio gen-eration (b) capabilities along training steps

tion on MMLU, GSM8K, and HumanEval, compared with Qwen2.5-7B-Instruct, which proves our model maintains strong common sense and text reasoning ability. Compared with the prior unified model in vision (Team, 2024; Wu et al., 2024) and pure speech (Tian et al., 2025a), our text-only metrics show a clear advantage. A noticeable observation during this training is that audio under-standing converges much faster than audio generation, which is evident in Fig.6.

---

[3]For CL, we enforce the winning-losing gap of 0.15; for (CE, CU, PC, PQ), we enforce that gap to be all positive. We only select one pair for each prompt.

Table 4: 5-scale subjective score of UALM-Reason on reasoning-oriented generation with 95% CI.

| Model | Enrichment | Dialogue | Self-reflection |
|---|---|---|---|
| UALM | $3.77 \pm 0.11$ | $3.92 \pm 0.11$ | $3.82 \pm 0.11$ |
| UALM-Reason | **4.01 ± 0.10** | **4.02 ± 0.10** | **4.04 ± 0.09** |

### 3.4  MULTIMODAL POST-TRAINING ANALYSIS

As the generative reasoning for audio models is still nascent, our evaluation mostly relies on qualitative analysis and subjective evaluation. We show that, with reasoning enabled, the UALM-Reason is superior than the base UALM.

**Qualitative Analysis:** Experimentally, our qualitative analysis shows that UALM-Reason excels in three key areas, compared with the base UALM. First, it demonstrates superior **detail controllability**, faithfully rendering nuanced acoustic details from text prompts that previous models find challenging. The model can discern concepts like number (*a dog barks vs. dogs bark*), spatial properties (*in the far distance*), temporal sequencing (*event A follows event B*), and audio texture (*distorted audio*)—a capability we attribute to our use of rich captions. Second, the model exhibits a sophisticated understanding of **human intention**. It successfully supports human-centric interactions like imaginary content *enrichment* and *dialogue* and subsequently generates appropriate intermediate rich captions. This is attributed to the base UALM's strong text capability, which connects its broad textual knowledge to the audio generation domain. Third, UALM-Reason creates a joint modeling scheme between understanding and generation. The model can analyze and criticize its own generated audio and leverage that critique for iterative refinement, an outcome of fine-tuning the base UALM for strong audio understanding. These observations are already beyond the existing evaluation protocols for audio generation. We thus demonstrate these experimental observations in our demo webpage.

**Subjective Evaluation:** Table 4 shows a 5-scale subjective evaluation for scenarios discussed in Section 2.4 (detail in Appendix C.3). The results demonstrate that generation-oriented reasoning from audio understanding and text capabilities is highly effective in advancing audio production.

## 4  RELATED WORK

**Language Model-Based Audio Generation:** Modern text-to-audio generation models can be broadly classified as diffusion models (Ziv et al., 2024; Liu et al., 2024b; Hung et al., 2024; Lee et al., 2024), auto-regressive language model (LM) approaches (Yang et al., 2023; Copet et al., 2024), and their hybrids (Lam et al., 2024). Recently, diffusion-based models are shown to have better generation quality (Hung et al., 2024; Lee et al., 2024). However, we demonstrate that with proper data scaling and a carefully designed training recipe (see §2.2), an LLM-based architecture can outperform diffusion models on this task. Furthermore, while conventional text-to-audio models (including both diffusion and LM approaches) often take contextual or contrastive embeddings (Chung et al., 2024; Wu et al., 2023) of the caption as cross-attention inputs, we show that a built-in BPE text tokenizer in a decoder-only LLM can achieve similar or better results.

While DPO (Rafailov et al., 2023) has been applied to diffusion-based text-to-audio generation (Hung et al., 2024; Majumder et al., 2024), this work is one of the first to integrate DPO into an LM-based text-to-audio framework and demonstrate its effectiveness. Both Hung et al. (2024); Majumder et al. (2024) and this work leverage the standard CLAP model for preference pair selection, though we acknowledge the potential benefits of adopting a human preference-aware CLAP (Takano et al., 2025) model to better align with human perception.

**Unified Audio Understanding and Generation:** There are many audio foundation models specialized in understanding (Kong et al., 2024; Goel et al., 2025; Chu et al., 2023; Xu et al., 2025) **or** generation (Yang et al., 2023; Liu et al., 2024b; Lee et al., 2024; Hung et al., 2024). To our knowledge, UALM is the first audio language model that successfully unifies these distinct capabilities within a single framework. Works with similar motivation have been explored in the vision and pure speech fields, but they often suffer from severe degradation on text-only tasks (Wu et al., 2024; Team, 2024; Tian et al., 2025a; Hori et al., 2019; Tjandra et al., 2017). In contrast, UALM

maintains high text reasoning capabilities, showing minimal degradation on text-only benchmarks during multimodal pre-training (Dai et al., 2024).

**Reasoning in Audio:** reasoning in existing audio language models is mostly for understanding and therefore confined to text-centric analysis of audio inputs, where the model connects acoustic cues to its pre-existing world knowledge (Xie et al., 2025; Diao et al., 2025; Goel et al., 2025; Rouditchenko et al., 2025; Deshmukh et al., 2025; Wu et al., 2025c). Reasoning in the generative domain is significantly under-explored. MusiCoT (Lam et al., 2025) is the most relevant recent work, which implements a chain-of-thought (CoT) process for music generation by predicting intermediate CLAP latents. By contrast, UALM-Reason carves out a new frontier by universally applying reasoning to understanding and generation – either separately or jointly. We demonstrate that reinforcement learning explored in prior understanding works (Diao et al., 2025; Rouditchenko et al., 2025) are also effective for reasoning beyond the text domain.

**Relationship with Speech Language Models (SLM):** The pre-trained UALM is designed for unified understanding and generation, a capability also explored in concurrent Speech Language Models (SLMs) (Défossez et al., 2024; Zeng et al., 2024). While these SLMs primarily target low-latency speech-to-speech dialogue and conversational smoothness, UALM targets at unified modeling of audio and text tasks in order to achieve higher-level audio intelligence. Furthermore, some SLMs employ joint modeling of text and audio within a single timestamp, relying on the strong assumption that speech and text are monotonically aligned. Since this assumption does not hold for general audio modeling, UALM treats text and audio as distinct, separate modalities.

## 5 CONCLUSION

We introduce the **U**nified **A**udio **L**anguage **M**odel (UALM), a single model that unifies audio understanding, text-to-audio generation, and text problem solving. We further present UALM-Reason, which leverages the understanding-generation together through novel multimodal reasoning capabilities – such as iterative refinement of its own outputs – a well-known behavior of high-level intelligence. This work marks a significant step towards more controllable, intelligent, and holistic audio AI. There are a number of important directions we plan to address in our future work.

**Unifying audio representation:** A key future direction of this work is to build a unified audio representation. As in §2.1, current UALM adopts the continuous audio encoder for audio input and the discrete audio codec tokens for audio output. A unified audio representation could further facilitate scalable training of joint understanding, generation, and reasoning.

**Quality Assessment of Synthetic Audio Captions:** Our SFT and DPO data curation is based on synthetic captions. However, even with our best efforts in data curation, there exists certain amount of hallucination and misalignment between the audio and captions through manual inspections. It is an important future direction to design quantitative methods at scale to assess the quality of the synthetic audio captions (especially the rich caption) to build a robust data curation pipeline.

**Quality Assessment of Audio Quality:** While there are many existing audio quality evaluation metrics (Lerch et al., 2025), there is still a gap from human perception. The lack of accurate and layered evaluation of acoustic quality, generation diversity, musical aesthetics and correctness, and faithfulness limits better RL of multimodal reasoning. In our future works, we plan to develop better audio quality evaluation metrics that are suitable for complex audio generation, including those measuring the multimodal reasoning chains.

REPRODUCIBILITY STATEMENT

To facilitate reproducibility, we thoroughly document our data and training procedures from the pre-training data mixture and recipes to the post-training curriculum for UALM-Reason. All training and inference hyper-parameters are provided in Appendix C.1. We also release the training code.

ETHICS STATEMENT

This research adheres to the ICLR Code of Ethics. Our model was trained on large-scale datasets, including standard benchmarks and audio with synthetically generated captions. We acknowledge that generative audio models carry a risk of misuse for creating misleading content. While this work is intended for creative and assistive applications, we encourage responsible use. The training data may reflect societal biases, which the model could perpetuate. We transparently note the use of LLMs for data curation and writing assistance in Appendix A.

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

## A  THE USE OF LLMS

We use LLMs to assist the writing of the paper in terms of: (1) grammar check, and (2) occasionally choosing the best word in writing. We use LLMs to verify that we have properly included all relevant papers in the field. We use LLMs as part of automated text data curation in our research, in a similar way as many other LLM-related research papers.

## B  TECHNICAL DETAILS

### B.1  RESIDUAL VECTOR QUANTIZATION AND DELAY PATTERN

For an audio frame $x \in \mathbb{R}^D$, RVQ discretizes $x$ into multiple discrete tokens by selecting one codebook per stage to successively quantize the residual. Concretely:

$$\hat{x} = \sum_{n=1}^{n_q} c_{n,i_n}, \quad i_n = \arg\min_k \|r_{n-1} - c_{n,k}\|, \ r_0 = x, \ r_n = r_{n-1} - c_{n,i_n}.$$

Here $n_q$ is the number of RVQ stages, $c_{n,k} \in \mathbb{R}^D$ is the $k$-th codebook at stage $n$, $i_n$ is the selected index, $r_n$ the post-stage residual, and $\hat{x}$ the reconstructed frame. For each $c_{n,k}$, we assign a unique discrete index $A_{n,k} \in \mathbb{N}$ to represent it.

Flattening all $n_q$ tokens per frame into a single sequence yields prohibitively long token streams ($T \times n_q$ steps for $T$ frames). To reduce effective audio sequence length, MusicGen (Copet et al., 2024) introduced the *delay* pattern: a single autoregressive Transformer predicts all $n_q$ codebooks in parallel at each step, with one LM head per codebook. As shown in Figure 7, the $n$-th codebook at frame $t - A_{n,t}$ – is predicted at sequence step $s = t + (n - 1)$, i.e. with a fixed temporal offset. Equivalently, at sequence step $s$, the LM predicts $n_q$ tokens $\{A_{1,s}, A_{2,s-1}, \cdots, A_{n_q,s-n_q+1}\}$ in parallel. The *delay* pattern allows the LM to capture dependencies across RVQ tokens while maintaining the required autoregressive sampling steps ($T + n_q - 1$) close to the number of audio frames ($T$).

In detail, the X-Codec (Ye et al., 2025) uses $n_q = 8$ and 50Hz frame rate ($T = 50 \times$ seconds). Therefore, for a 10-second audio, there are 4000 tokens to be predicted in 507 sequence steps.

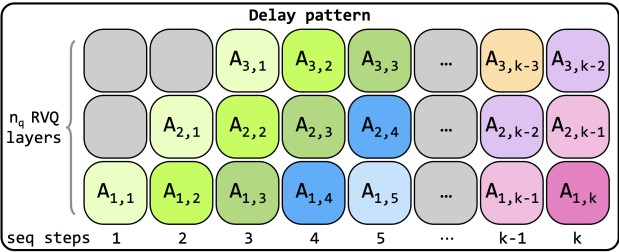

Figure 7: Illustration of *delay* pattern of RVQ audio tokens $A$. $A_{n,k}$ is the $k$-th audio token at $n$-th RVQ layer, $1 \leq n \leq n_q$, $1 \leq k \leq T$. At step $s$, the LM predicts $\{A_{1,s}, A_{2,s-1}, \cdots, A_{n_q,s-n_q+1}\}$ in parallel.

### B.2  ENHANCEMENT VAE

UALM operates in 16kHz monophonic audio waveform, following previous works in continuous encoder (Goel et al., 2025) and discrete codec (Ye et al., 2025). To improve perceptual quality of generated audio, we attach an enhancement module based on VAE (Kingma & Welling, 2014). It receives 16kHz mono waveform generated by the audio codec decoder and upsamples into 48kHz stereo waveform. Note that the input waveform to the enhancement VAE is a lossy version of the target with codec distortions; the enhancement VAE implicitly learns the joint process of blind audio restoration, super-sampling, and spatialization.

The encoder consists of an STFT layer (`hop_length=8`) that converts the waveform to spectrogram, followed by 3 ConvNeXt (Liu et al., 2022)-based downsampling blocks (`stride=[4, 4, 5]`) with 153M parameters (starting with 512 channels, multiplied by 2x per block) similar to Vocos (Siuzdak, 2023). Thus, the encoder downsamples the input by 640x producing latents at 25Hz frame rate. The latent dimension is set to 512. The decoder consists of 5 convolutional upsampling blocks (`stride=[8, 6, 5, 4, 2]`) with 489M parameters (starting with 2560 channels) based on Stable Audio Open (Evans et al., 2024), upsampling the latent by 1920x to generate 48kHz stereo waveform. All layers use *SnakeBeta* activation functions (Lee et al., 2023).

The enhancement VAE is trained with a combination of reconstruction, adversarial, and regularization losses. Let $x_{\text{in}}$ denote the distorted waveform, and $x_{\text{gt}}$ the corresponding clean ground-truth waveform. The encoder $E$ maps $x_{\text{in}}$ to a latent distribution $q_E(z|x_{\text{in}})$, and the decoder $D$ reconstructs $\hat{x} = D(z)$ from a latent sample $z \sim q_E$. The enhancement VAE is trained with the following objective:

$$L_{\text{VAE}} = L_{\text{stereoMRSTFT}} + L_{\text{logmel}} + L_{\text{adv}} + L_{\text{feat}} + \zeta \cdot L_{\text{KL}}, \tag{3}$$

where each loss term is defined as follows:

- Stereo sum and difference MR-STFT loss (Steinmetz & Reiss, 2020; Steinmetz et al., 2021)

$$L_{\text{MRSTFT}}(x, \hat{x}) = \sum_{i=1}^{m} \left( \frac{\|\text{STFT}_i(x) - \text{STFT}_i(\hat{x})\|_F}{\|\text{STFT}_i(x)\|_F} + \frac{1}{T} \left\| \log \frac{\text{STFT}_i(x)}{\text{STFT}_i(\hat{x})} \right\|_1 \right), \tag{4}$$

$$L_{\text{stereoMRSTFT}}(x_{\text{gt}}, \hat{x}) = L_{\text{MRSTFT}}(x_{\text{sum}}, \hat{x}_{\text{sum}}) + L_{\text{MRSTFT}}(x_{\text{diff}}, \hat{x}_{\text{diff}}), \tag{5}$$

  with $x_{\text{sum}} = x_{\text{gt,L}} + x_{\text{gt,R}}$ and $x_{\text{diff}} = x_{\text{gt,L}} - x_{\text{gt,R}}$.
- Multi-scale log-mel L1 loss (Kumar et al., 2023)

$$L_{\text{logmel}}(x_{\text{gt}}, \hat{x}) = \sum_{j=1}^{J} \left\| \log_{10}\big(\text{mel}_j(x_{\text{gt}})\big) - \log_{10}\big(\text{mel}_j(\hat{x})\big) \right\|_1, \tag{6}$$

  where $\text{mel}_j$ denotes a mel spectrogram at resolution $j$.
- Least-squares adversarial loss (Mao et al., 2017) with stereo BigVGAN-v2 discriminator (Lee et al., 2023)

$$L_{\text{adv}}(x_{\text{gt}}, \hat{x}) = \frac{1}{K} \sum_{k=1}^{K} \left[ (D_k(x_{\text{gt}}) - 1)^2 + (D_k(\hat{x}))^2 \right], \tag{7}$$

  where $D_k$ is the $k$-th discriminator head.
- Feature matching L1 loss (Larsen et al., 2016)

$$L_{\text{feat}}(x_{\text{gt}}, \hat{x}) = \frac{1}{KL} \sum_{k=1}^{K} \sum_{l=1}^{L} \frac{\|D_k^l(x_{\text{gt}}) - D_k^l(\hat{x})\|_1}{\text{mean}(\|D_k^l(x_{\text{gt}})\|_1)}, \tag{8}$$

  where $D_k^l(\cdot)$ is the $l$-th feature map of discriminator $D_k$.
- KL divergence regularization

$$L_{\text{KL}} = \text{KL}(q_E(z|x_{\text{in}}) \,\|\, \mathcal{N}(0, I)). \tag{9}$$

## C  EXPERIMENTAL DETAILS

### C.1  TRAINING AND INFERENCE HYPER-PARAMETERS

Table 5: Pre-Training configurations

|  | Modality Alignment | Pre-Training |
|---|---|---|
| Batch Size (Token per GPU) | 25k | 5k |
| Peak Learning Rate | 5e-4 | 1e-4 |
| Learning Schedule | Constant | Cosine Decay |
| Warmup Steps | 0 | 25k |
| #Steps | 1.8k | 660k |
| #GPUs | 128 | 128 |

Table 6: Post-Training configurations

|  | SFT1 | DPO1 | SFT2 | DPO2 |
|---|---|---|---|---|
| Batch Size (Token per GPU) | 5k | 3k | 5k | 3k |
| Peak Learning Rate | 2e-6 | 2e-7 | 2e-7 | 2e-7 |
| Learning Schedule | Constant | Constant | Constant | Constant |
| Warmup Steps | 0 | 0 | 0 | 0 |
| #Steps | 15k | 2k | 15k | 2k |
| #GPUs | 32 | 8 | 32 | 8 |
| #Samples | 750k | 60k | 850k | 20k |
| DPO $\beta$ | - | 0.1 | - | 0.1 |
| DPO Cross-Entropy Weight | - | 1.0 | - | 1.0 |

Table 7: Inference Configuration

| Modality | Method | Candidate $k$ | CFG $\lambda$ | Temperature |
|---|---|---|---|---|
| Text | Greedy Search | - | - | - |
| Audio | Top-k Sampling | 20 | 3.0 | 1.0 |

## C.2  AUDIO GENERATION: ABLATION STUDY

Table 8: Ablation study of UALM-Gen and UALM showing the effect of CFG, DPO, and Enhancement VAE. Objective metrics include: (1) Frechet distance (FD) using OpenL3 (Cramer et al., 2019) at 44.1kHz; (2) Kullback–Leibler divergence (KL) using PaSST (Koutini et al., 2022) at 32kHz; (3) Inception Score (IS) using PANNs (Kong et al., 2020) at 16kHz; (4) CLAP scores (CL) using LAION-CLAP (Wu et al., 2023) at 48kHz; (5) AudioBox-Aesthetic score (AES) (Tjandra et al., 2025) using an average of (CE, CU, PC, PQ) at 16kHz.

| Model | SR (kHz) | SongDescriber FD ↓ | KL ↓ | IS ↑ | CL ↑ | AES ↑ | AudioCaps FD ↓ | KL ↓ | IS ↑ | CL ↑ | AES ↑ |
|---|---|---|---|---|---|---|---|---|---|---|---|
| Ground Truth | – | 0 | 0 | 1.88 | 0.48 | 7.20 | 0 | 0 | 13.49 | 0.62 | 4.50 |
| Ground Truth (X-codec reconstructed 16kHz) | 16 | 183.99 | 0.16 | 1.68 | 0.45 | 7.27 | 172.74 | 0.58 | 8.29 | 0.45 | 4.41 |
| MusicGen-stereo-L (Copet et al., 2024) | 32 | 228.94 | 0.84 | 1.69 | 0.36 | 6.64 | — | — | — | — | — |
| AudioGen-M (Kreuk et al., 2022) | 16 | — | — | — | — | — | 149.70 | 1.35 | 12.26 | 0.57 | 4.21 |
| MAGNeT-M (Ziv et al., 2024) | 32 | 191.49 | 0.70 | 1.47 | 0.41 | 6.65 | 149.68 | 1.86 | 7.73 | 0.46 | 4.08 |
| AudioLDM2-L (Liu et al., 2024b) | 16 | 331.73 | 0.68 | 1.96 | 0.45 | 6.31 | 121.63 | 1.72 | 8.59 | 0.52 | 4.31 |
| Make-An-Audio (Huang et al., 2023) | 16 | 277.00 | 1.85 | 2.68 | 0.28 | 5.44 | 189.28 | 1.51 | 10.31 | 0.54 | 4.30 |
| AudioLCM (Liu et al., 2024c) | 16 | 261.11 | 1.31 | 2.24 | 0.33 | 5.65 | 166.20 | 1.49 | 11.06 | 0.52 | 4.32 |
| TangoFlux (Hung et al., 2024) | 44.1 | 235.61 | 0.71 | 1.70 | 0.41 | 6.46 | 103.04 | **1.02** | 15.13 | **0.65** | 4.42 |
| Stable Audio Open (Evans et al., 2024) | 44.1 | 138.58 | 1.01 | **2.25** | 0.42 | 6.37 | 100.93 | 2.22 | 11.80 | 0.35 | 4.47 |
| ETTA (Lee et al., 2024) | 44.1 | 95.66 | 0.80 | 2.15 | 0.44 | 6.71 | 80.13 | 1.22 | 14.36 | 0.54 | 4.51 |
| UALM-Gen-Base (w/o CFG) | 16 | 232.21 | 1.06 | 2.11 | 0.39 | 6.51 | 186.68 | 3.00 | 5.46 | 0.25 | 4.30 |
| UALM-Gen-Base | 16 | 217.90 | 0.84 | **2.13** | 0.45 | 6.70 | 186.01 | 1.23 | 10.86 | 0.51 | 4.47 |
| + DPO | 16 | 224.72 | 0.68 | 1.85 | 0.51 | **7.36** | 214.89 | **1.16** | 13.43 | 0.57 | 4.99 |
| + Enhancement VAE (UALM-Gen) | 48 | **74.43** | 0.63 | 1.87 | **0.54** | **7.36** | 75.14 | 1.19 | 14.52 | **0.65** | **5.08** |
| UALM-Base | 16 | 212.78 | 0.73 | 2.05 | 0.47 | 6.82 | 181.01 | 1.59 | 10.48 | 0.45 | 4.40 |
| + DPO | 16 | 207.82 | 0.67 | 1.96 | 0.52 | 7.28 | 196.56 | 1.25 | 14.36 | 0.53 | 4.87 |
| + Enhancement VAE (UALM) | 48 | 83.69 | **0.59** | 2.00 | **0.54** | 7.28 | **65.87** | 1.35 | **15.62** | 0.62 | 4.92 |

Table 8 shows an ablation study of UALM-Gen and UALM. We note the ablation model without DPO and the enhancement VAE module using a '-Base' suffix. First, activating CFG significantly improves prompt adherence measured by CL, along with improving all other objective metrics. Applying DPO provides further improvements, especially CL and AES. Finally, applying enhancement VAE gives significant improvements in FD along with overall improvements of other metrics.

### C.3 DETAILS ON SUBJECTIVE EVALUATION OF REASONING-BASED GENERATION

We conducted a 5-scale mean opinion score analysis using Mechanical Turk for three reasoning-oriented generation scenarios described in Section 2.4. We uniformly curate 20 test prompts/dialogues for each of the below categories, with instructions for evaluators as follows:

**Enrichment** tests the model's ability to interpret creative user requests, where evaluators rated how well generated audio fulfilled imaginative descriptions (e.g., "I need an audio track for a club scene - something energetic and modern").

---

**User Request:** I need a soundscape that evokes a sense of intense cleaning. Think of someone trying to get something really dirty, sparkling clean!

> ▶ 0:00 / 0:10  ——   🔊  ⋮

**REL (Relevance): How well does the audio fulfill the user's creative request?**

○ 5: Excellent - Audio perfectly fulfills the user's request and captures all described elements
○ 4: Good - Audio mostly fulfills the user's request with minor elements missing
○ 3: Fair - Audio somewhat matches the user's request but lacks important elements
○ 2: Poor - Audio barely relates to what the user requested
○ 1: Bad - Audio does not match the user's request at all

---

**Dialogue** evaluates the model's capacity to handle multi-turn conversations, where users iteratively refine their audio requirements with an assistant. Evaluators judge whether the final audio matched the user's accumulated specifications based on the dialogue.

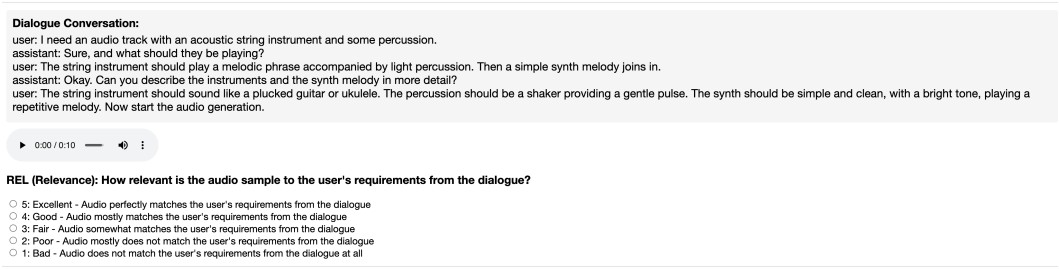

**Dialogue Conversation:**
user: I need an audio track with an acoustic string instrument and some percussion.
assistant: Sure, and what should they be playing?
user: The string instrument should play a melodic phrase accompanied by light percussion. Then a simple synth melody joins in.
assistant: Okay. Can you describe the instruments and the synth melody in more detail?
user: The string instrument should sound like a plucked guitar or ukulele. The percussion should be a shaker providing a gentle pulse. The synth should be simple and clean, with a bright tone, playing a repetitive melody. Now start the audio generation.

> ▶ 0:00 / 0:10  ——   🔊  ⋮

**REL (Relevance): How relevant is the audio sample to the user's requirements from the dialogue?**

○ 5: Excellent - Audio perfectly matches the user's requirements from the dialogue
○ 4: Good - Audio mostly matches the user's requirements from the dialogue
○ 3: Fair - Audio somewhat matches the user's requirements from the dialogue
○ 2: Poor - Audio mostly does not match the user's requirements from the dialogue
○ 1: Bad - Audio does not match the user's requirements from the dialogue at all

---

**Self-Reflection** checks event correctness, focusing on semantic accuracy, requiring evaluators to assess whether all specified audio events were present and correctly ordered in the generated output.

---

**Text Description:** Chaotic and rapid ringing of large metallic bells.

**Remember:** Rate ONLY whether events are correct, IGNORE audio quality/artifacts

> ▶ 0:00 / 0:10  ——   🔊  ⋮

**CORRECTNESS (Event Correctness): Are all the audio events mentioned in the text present and correct? (IGNORE audio quality/artifacts)**

○ 5: Perfect - ALL events from the text are present and correct (ignore audio quality)
○ 4: Good - MOST events are present and correct, minor events might be missing
○ 3: Fair - SOME important events are present, but several are missing or wrong
○ 2: Poor - FEW events are correct, most are missing or wrong
○ 1: Bad - Events are completely wrong or missing

---

## D    DATA CURATION DETAILS

This appendix details the data curation strategy employed for constructing our text-to-audio generation training pairs.

**Data Sources:** We collect audio data and annotations from existing works including: Stable-Audio-Open (Evans et al., 2024), ETTA (Lee et al., 2024), Audio Flamingo 3 (Goel et al., 2025), and AudioSetCaps (Bai et al., 2025), following the data preparation instructions of each of these works. For text-only data, we use corpus from Liu et al. (2025) and Bercovich et al. (2025a).

We use the full audio understanding data and text data from these sources. We only conduct synthetic data labeling and data filtering for text-to-audio data as follows.

**Labeling:** For audio without a provided caption, we segment them into non-overlapping 10-second segments. We use open-sourced captioning models including Qwen-2.5-Omni (Xu et al., 2025) and Audio Flamingo 3 (Goel et al., 2025) to generate multiple captions for each clip. These models were prompted with the instruction: *"Generate audio caption for the input audio."*

**Data Filtering:** To ensure high data and caption quality, we improve the filtering pipeline of ETTA (Lee et al., 2024) as follows:

1. **De-duplication:** We compute AF-CLAP (Goel et al., 2025) audio embeddings and remove repetitive audio using FAISS-GPU.

2. **Quality Filtering:** We identify repetitive, low-quality keywords, and low-semantic-content keywords to remove low quality audio and captions. The full keywords are: `["noise", "noisy", "unclear", "muffled", "indistinct", "inaudible", "distorted", "garbled", "unintelligible", "static", "interference", "echo", "background noise", "low volume", "choppy", "feedback", "crackling", "hissing", "fuzzy", "murmur", "buzzing", "scrambled", "faint", "broken up", "skipped", "irrelevant", "overlapping speech", "reverberation", "clipping", "sibilance", "popping", "unspecific", "gibberish", "unknown sounds", "vague", "ambiguous", "incoherent", "misheard", "uncertain", "distant", "irregular", "glitch", "skipping", "dropout", "artifact", "undermodulated", "overmodulated", "off-mic", "misinterpretation", "unreliable", "fluctuating", "low-quality", "low quality", "compromised", "substandard", "inferior", "deficient", "poor", "suboptimal", "flawed", "unsatisfactory", "inadequate", "faulty", "second-rate", "mediocre", "insufficient", "lacking", "imprecise"].`

3. **Audio-Text Alignment:** We calculated the AF-CLAP (Goel et al., 2025) audio-text similarities for each sample pair and set a threshold of 0.25. We note that AF-CLAP is only used for filtering and is different from the CLAP model used for text-to-audio evaluation.

