# OpenReview forum: "UALM: Unified Audio Language Model for Understanding, Generation and Reasoning"
_ICLR.cc/2026/Conference — ICLR 2026 Oral_

### Official Review · Reviewer_tZVa · 2025-10-25

**Soundness:** 4
**Presentation:** 4
**Contribution:** 4
**Rating:** 8
**Confidence:** 4

**Summary:**

This paper introduces UALM : a joint LLM model for audio understanding, text-to-audio generation, and text reasoning.
They train an audio-reasoning model UALM-R1 that has the ability to correct its own generation. They show that LLMs can beat diffusion models at text-to-audio generation, but have different scaling laws and require much more data.

Main contributions:
- They introduce a very carefully crafted training recipe, with modality alignment, multi-task pre-training, SFT and DPO. Showing that LLM based audio gen models need 10x data compared to diffusion audio gen models.
- Unified understanding + generation model with state of the art results on text to audio generation- the authors plan to open-source model weights upon publication
- First attempt at reasoning for audio generation (UALM-R1) with decent increase in performance compared to UALM

**Strengths:**

- State of the art results on text-to-audio generation with a LLM, whereas all previous work pointed towards diffusion models being the strongest models for text-to-audio. The authors show that with ~10X more data, LLMs can outperform diffusion models. This is a very promising direction for future research around audio-gen LLMs and scaling laws
- A very well engineered pipeline for model training and post-training - with details clearly laid out in the paper.
- Ablation studies of newly introduced parameters : CFG for LLM sampling, self-adaptation before DPO,  cross entropy loss regularization for DPO
- Comparison to previous state of the art audio-generation models on established baselines
- Supplemental material shows audio generation of excellent quality + comparison to state of the art models for the same prompts, very well documented

**Weaknesses:**

- The reasoning model UALM-R1 is only evaluated on "instruction following" tasks but its audio quality isn't compared directly with audio metrics to the base model. Reasoning is done mostly through this rich caption text abstraction but there's no "soft" signal extracted from the generation to improve it.
- The authors do not show synergies between audio-understanding and audio-generation joint training - the synergy is exploited for the reasoning model but they don't discuss if audio-understanding improves audio-generation performance or the other way around.

**Questions:**

- For DPO, you mention the cross entropy loss is necessary to adapt to generation. Do you adapt to generations (generated with CFG) from the CFG=1 model? How does this regularization compare to using different DPO beta to stick closer to base policy?
- For the RVQ and delay pattern appendix - you describe quite clearly how this works at the output layer, how does this work at the input layer?
- For the enhancement VAE - why use a VAE instead of e.g a GAN or diffusion model to upsample and correct the audio? This looks like a key step in getting SOTA FD metric

---

> ### Author Response · Authors · 2025-11-21
> **Reply to reviewer**
>
> We appreciate your positive feedback and address your questions below.
>
> - Weakness 1 (UALM-R1 evaluation)
>   - We thank the reviewer for this question. We would like to clarify that our subjective evaluation indicates the UALM-R1 outperforms UALM overall, which already takes audio quality and instruction-following into consideration. Text-to-audio metrics are not suitable for this because the context can be very complicated especially in the dialogue and self-reflection paradigms.
>   - We would like to clarify that in our self-reflection paradigm, UALM-R1 directly takes its first-round generation as audio inputs and refines it. This could be considered as the "soft" signal being usage.
>
> - Weakness 2 (Understanding-Generation Synergies)
>   - We appreciate this comment, and we would like to clarify that our claim is: *Concurrent generation and understanding capabilities within one single model is necessary to unlock high-level intelligence in audio*, and we do not intend to show in this work that *joint training on understanding and generation tasks could directly facilitate each other under existing evaluation scheme*, which we aim to achieve in our future work. Per your suggestion, we have replaced the word **synergy** with **"a joint modeling scheme"** to avoid ambiguity.
>   - We would like to claim that the **joint modeling scheme** enables critical capabilities that were impossible in single-task models. For instance, the "Self-Reflection" mechanism (Section 2.4) allows the model to generate, listen, think, and then revise. We aim to show that with our unified pre-training, one can achieve any interleaved audio-text format as long as such post-training data are curated.
>
> - Q1 (DPO details)
>   - For the initial adaptation stage, we utilize self-generated samples produced with CFG = 3. We chose this value to match our inference setup, ensuring the model adapts to the specific distribution of high-quality outputs it is expected to generate.
>   - Our early experiments indicate that this supervised adaptation step is necessary for DPO, independent of hyperparameter tuning. If we skip it, even when we use a high DPO $\beta$ or add concurrent cross-entropy regularization, we consistently observe performance degradation (though higher $\beta$ leads to slightly less degradation). We have revised Section 2.2 to clarify these implementation details.
>
> - Q2 (Delay pattern at input layer)
>   - We apply the delay pattern symmetrically. During training, the input audio tokens (which serve as the autoregressive history) are interleaved in the exact same manner as the output targets. This ensures that the model rigorously adheres to the next-token-prediction objective in a multi-stream fashion, maintaining temporal alignment across all codebook layers.
>
> - Q3 (Enhancement VAE)
>   - We aim to train a model that takes X-Codec reconstructed (degraded) audio as input and outputs high-quality audio with less artifacts and wider bandwidth. VAE is suitable for this task and we combine reconstruction and GAN losses when training it. We find diffusion models work similarly well but they are much ($>1000\times$) slower at inference time, thus not suitable for our task.
>   - We would like to clarify that the Enhancement VAE only fixes X-Codec artifacts and degradations and does not generate any novel data. All the semantic contents and novel sounds are generated via UALM core component. Since X-Codec's original decoder produces bad acoustic quality, it is natural to append a separate, light-weight post-processing model for acoustic improvements. This Enhancement VAE can also be regarded as an improved X-Codec decoder, which we will clarify in the paper. The FD (OpenL3) evaluates 48kHz stereo audio so it really punishes 16kHz outputs; other 16kHz metrics are more or less similar.

---

### Official Review · Reviewer_9tTC · 2025-10-30

**Soundness:** 2
**Presentation:** 3
**Contribution:** 3
**Rating:** 2
**Confidence:** 4

**Summary:**

This paper introduces UALM, a single AI model that unifies three audio tasks: understanding, generation, and reasoning. Instead of using separate models, UALM uses one large language model to handle everything. The special version, UALM-R1, pioneers multimodal reasoning by using intermediate "rich captions" (detailed audio plans) and a "self-reflection" capability, where it generates an audio clip, "listens" to its own output, critiques flaws, and then generates an improved version.

**Strengths:**

1.This paper uses one model to do three jobs: audio understanding, audio generation, and reasoning. This is simpler and more efficient than making a separate model for each task.

2.The paper introduces a "self-reflection" feature. The model can "listen" to the audio it generated, find what's wrong, and then make a better version. This helps improve the final audio quality and accuracy.

**Weaknesses:**

1.The comparison in many places such as Table 2 (Audio Understanding) lacks rigor. The paper does not list the specific parameter counts for competing models like SALMONN or Step-audio-2. These models often have multiple versions of different sizes (e.g., 1B, 7B, 72B). Without this, we cannot know if the comparison against UALM is fair, making the results inconclusive.

2.The paper's most novel feature, "self-reflection" (UALM-R1), lacks strong evidence. Its evaluation (Table 4 & Appendix C.3) relies only on a tiny subjective test (20 prompts per scenario), which is not enough to prove it works reliably. Furthermore, in the demos, the "improvement" sounds like it just adds new audio on top of the original, rather than truly "correcting" or "replacing" the mistake.

3.Limited innovation and poor reproducibility. The paper's approach relies heavily on combining existing techniques (like the Encoder-Adapter-LLM architecture and DPO) rather than introducing new methods. Its strong performance seems to come from brute-forcing the training with a massive 30 million sample (80k hours) dataset. The paper does not detail how this dataset was collected or created, and the dataset is private. This makes the work difficult to reproduce, and it's unclear if the success is due to the model's design or just the enormous amount of private data.

**Questions:**

1.In Table 2 (Audio Understanding), the paper did not provide the parameter counts for competitor models (like Audio Flamingo 3 or Qwen2.5-Omni). Can the paper provide these numbers?

2.The reasoning (SFT-1/SFT-2) and generation (30M) data rely heavily on synthetic creation or pseudo-labels. Could the author detail the exact data generation pipeline for reproducibility? For example, what models and prompts were used to generate the SFT data, and how were the initial 250k "rich captions" created?

---

> ### Author Response · Authors · 2025-11-21
> **Reply to reviewer**
>
> We appreciate your valuable comments and address your questions below.
>
> - Weakness 1 and Q1 (model size)
>   - We thank the reviewer for this comment. To ensure rigor and transparency, we have updated Table 1 to explicitly list the parameter counts for all competing models. Our base LLM is the smallest (7B) among all baselines. On MMAU, our model even outperforms the potentially very large Step-Audio-2 (their mini version has 7B parameters and the full model is described to have "few than 130B parameters".)
>   - [1] Wu, Boyong, et al. "Step-audio 2 technical report." arXiv preprint arXiv:2507.16632 (2025).
>
> - Weakness 2 (UALM-R1 evaluation)
>   - Our contribution of multimodal reasoning (self-reflection, enrichment, and dialogue) mostly conveys the idea that **with unified modeling across text and audio, one could break the limit of conventional text-only reasoning and achieve flexible interleaved audio-text capabilities** -- as long as one could curate such data pairs, which becomes an engineering or data annotation problem. Our demos have shown this possibility that was never achieved in the literature before, and direct a feasible path towards more advanced audio general intelligence beyond what current (e.g. diffusion based) models are able to achieve.
>   - We would like to note that there lack evaluation protocols for these complex abilities, so we end up with human evaluation, which is costly and not very scalable. Nevertheless, we will take your suggestion seriously and conduct more rigorous evaluation regarding different specific skills in our future work.
>   - We also kindly note that the error correction in self-reflection contains multiple forms. As suggested by the reviewer, new audio events can be added when they are detected as missing, which is the ideal behavior. E.g., in the 5th demo, the model correctly identifies that the "car engine sound" is missing and generates a new version containing that sound. In other cases, the correction can behave differently from adding an audio event. E.g., in the 3rd demo, the model critiques a clip containing "multiple insects" and corrects it to "a single insect", the evidence of replacement.
>
> - W3 (innovation and reproducibility)
>   - We respectively disagree with this comment. The data to train UALM-Gen and UALM are completely public, and we detail the sources in Appendix E. We list our contributions and innovations below.
>     - Current SOTA text-to-audio models are mostly diffusion-based. Our contribution with respect to text-to-audio is: **with proper data scaling, classifier-free guidance in LLM, and proper DPO, the LLM-based text-to-audio model could match SOTA diffusion models.**
>     - Our contribution with respect to the unified model is: **a single model could integrate audio understanding, audio generation, and text reasoning, and can match the experts of each task.**
>     - Our contribution with respect to multimodal reasoning is: **we break the boundary between audio and text modalities and thus can achieve flexible interleaved audio-text outputs as long as we provide data in such formats** (which becomes a data annotation problem).
>   - These are novel findings in the research community and point a steady path towards more advanced audio general intelligence.
>   - We disagree that our efforts on data scaling are interpreted as "brute-force". As in Fig 5b, we clearly demonstrate the impact of data scaling, which highlights the extreme importance of data scaling, providing a reliable direction to optimize LLM-based text-to-audio models. Besides, we also provide key insights on combining data from different modalities, training with classifier-free guidance, principled training stages, and many other valuable techniques.
>
> - Q2 (data)
>   - We describe the details of UALM-Gen and UALM data in Appendix E, which is ready for reproduction. The rich captions are internally available, which could be reproduced or even improved by human annotation or running latest multimodal LLMs.

---

### Official Review · Reviewer_jfVq · 2025-10-30

**Soundness:** 2
**Presentation:** 2
**Contribution:** 4
**Rating:** 8
**Confidence:** 4

**Summary:**

This paper presents UALM, a unified audio-language model designed for understanding, generation, and reasoning. The authors further introduce UALM-R1, which demonstrates enhanced reasoning capabilities. Overall, the paper is clearly written and easy to follow.

**Strengths:**

- The paper is clearly written and well organized. The accompanying demo page provides convincing qualitative results.

- The three main challenges are well defined and clearly articulated.

- The proposed methodology—pre-extracting audio tokens and continuing the training of a pretrained LLM—appears effective and contributes to the model’s success.

- The paper provides several insightful observations, such as the comparable data requirements between LLMs and diffusion models, and the use of classifier-free guidance. These are valuable takeaways for the research community.

**Weaknesses:**

- The main concern lies in the dataset preparation and processing, which currently lack sufficient detail. I would be like to accept this paper if the dataset construction process is clearly described.

- The paper should provide detailed information on how the 30M text–audio pairs were created, including data sources, filtering, and preprocessing steps.

- In Equation (1), the meaning of the symbol \pi is unclear — does it denote a probability distribution?

- The reproducibility of the research is uncertain, as important implementation and dataset details are not provided.

**Questions:**

N.A.

---

> ### Author Response · Authors · 2025-11-21
> **Reply to reviewer**
>
> We appreciate your positive feedback.
>
> We address your Q1, Q2 and Q4 below:
> - We provide more details about our data preparation and processing in Appendix E of the updated manuscript, including the data sources, labeling and filtering strategies. We believe these details improve the transparency and reproducibility of our work.
>
> As for your Q3
> - Yes, $\pi$ denotes a probability distribution: $\pi_{\theta}$ represents the policy of the language model with trainable parameters $\theta$. It outputs the conditional probability distribution over the next token, formulated as $\pi_{\theta}(y_t \mid y_{<t}, x)$. We use the notation $\pi$ to be consistent with Equation (2) following the original Direct Preference Optimization (DPO) paper.}
>
> [1] Rafailov, Rafael, et al. "Direct preference optimization: Your language model is secretly a reward model." Advances in neural information processing systems 36 (2023): 53728-53741.

---

### Official Review · Reviewer_jHYo · 2025-11-01

**Soundness:** 3
**Presentation:** 3
**Contribution:** 3
**Rating:** 6
**Confidence:** 2

**Summary:**

The paper proposes UALM, a single decoder only LLM (initialized from Qwen2.5 7B) that is extended with an audio encoder + MLP adapter for inputs and an audio codec head for outputs. It aims to unify three capabilities in one model: (i) audio understanding, (ii) text to audio generation, and (iii) multimodal reasoning that interleaves text and audio in intermediate steps. Empirically, UALM matches or outperforms strong baselines on AudioCaps and SongDescriber.

**Strengths:**

S1.
The design choices (25 Hz input frames, 50 Hz codec frames with 8 RVQ codes, loss scaling so one audio frame ≈ one text token) are well motivated and concrete.

S2.
UALM Gen and UALM deliver competitive objective scores vs. leading diffusion models on both SongDescriber and AudioCaps.

S3.
The rich caption representation (keywords, temporal layout, and detailed descriptions) plus enrichment, dialogue, and self reflection workflows are well illustrated.

**Weaknesses:**

W1.
The generation corpus is largely pseudo labeled, and DPO pair selection uses CLAP + aesthetics metrics. This raises a risk that the model (and the enhancement VAE) overfit to the judge/feature space that also underlies the reported objective scores.

W2.
UALM R1’s core claim is generation oriented reasoning, yet the subjective study uses only 20 prompts per scenario with AMT raters, and there is no standardized benchmark for audio CoT controllability (e.g., temporal ordering, count, spatial positioning) beyond demos.

W3.
The alignment warm up is asserted to be critical, but there’s no ablation isolating its contribution vs. simply training longer/fewer steps or different adapter capacity.

W4.
Weights/code are promised “upon publication", but several key assets (the 30M corpus, rich caption pairs, selection heuristics for DPO, and the enhancement VAE checkpoints) are not available. Given the centrality of the enhancement VAE and the data scale, releasing exact configs and checkpoints (even if data cannot be released) is important.

**Questions:**

Please refer to the weaknesses above.

---

> ### Author Response · Authors · 2025-11-21
> **Reply to the reviewer**
>
> - W1 (overfitting to objective metrics)
>   - We thank the reviewer for bringing up this important question and we would like to clarify as follows. First, we use AF-CLAP [1] for DPO pair selection and LAION-CLAP [2] for evaluation; these are distinct models. Second, while AES is used in both DPO pair selection and evaluation, the effectiveness of AES-based DPO is justified by the improvements in other subjective and objective evaluation metrics and also prior works [3]. Third, the caption labeling models are trained via autoregressive LLM and they do not directly optimize our reported objective metrics. Last, the Enhancement VAE aims to fix X-Codec degradations (because X-Codec decoded audio contains artifacts and sounds quite bad); it is not trained on any of the objective metrics.
>   - [1] Ghosh, Sreyan, et al. Audio flamingo 2: An audio-language model with long-audio understanding and expert reasoning abilities.
>   - [2] Wu, Yusong et al. Large-scale Contrastive Language-Audio Pretraining with Feature Fusion and Keyword-to-Caption Augmentation.
>   - [3] Chia-Yu Hung, et al, TangoFlux: Super fast and faithful text to audio generation with flow matching and clap-ranked preference optimization.
>
> - W2 (UALM R1 evaluation)
>   - We appreciate the reviewer for pointing this out. This is a highly under-explored area -- no evaluation protocols, no benchmarks, and even no training data; our motivation is to call the research community's attention and show initial results on what is feasible under our framework. Due to lack of evaluation protocols, we conduct human evaluation but it is costly and not scalable by nature. We will take your advice seriously and investigate in more depth the evaluation of advanced multimodal reasoning abilities with specific skills as you mentioned.
>
> - W3 (warmup ablation)
>   - We appreciate you ask this important question. We experimented without this warm-up stage: the text reasoning ability has a huge degradation as training proceeds (see the table below). This justifies the necessity of the warm-up stage. Also, this warm-up stage is a standard practice in building state-of-the-art multimodal LLMs [4, 5], including both audio and vision domains.
>   - [4] Goel, Arushi, et al. "Audio flamingo 3: Advancing audio intelligence with fully open large audio language models."
>   - [5] Wu, Chengyue, et al. "Janus: Decoupling visual encoding for unified multimodal understanding and generation"
> -----
> | Steps (No Warm-up) | 20k | 40k | 60k |
> | :--- | :---: | :---: | :---: |
> | GSM8K | 78.9 | 74.6 | 72.0 |
> | HumanEval | 61.6 | 56.7 | 45.1 |
> -----
>
> - W4 (key assets)
>   - We will facilitate transparent and reproducible research through open-sourcing. We have described more details of our data in Appendix E.

---

### Official Review · Reviewer_Gqug · 2025-11-03

**Soundness:** 3
**Presentation:** 3
**Contribution:** 3
**Rating:** 6
**Confidence:** 4

**Summary:**

ALM), an ambitious framework designed to unify audio understanding, text-to-audio generation, and multimodal reasoning within a single model. The work presents a three-stage approach: (1) It first develops UALM-Gen, a language model-based text-to-audio generator that, through extensive data scaling and techniques like Classifier-Free Guidance (CFG) and Direct Preference Optimization (DPO), achieves quality comparable to state-of-the-art diffusion models. (2) It then presents UALM, a single model trained via a carefully designed data blending and curriculum learning strategy to concurrently handle audio understanding, generation, and text-only reasoning tasks without significant performance degradation. (3) Finally, it introduces UALM-R1, a reasoning-enhanced model post-trained to perform novel multimodal generative reasoning tasks, such as generating audio from abstract prompts (Enrichment), interactive refinement (Dialogue), and iterative self-correction (Self-Reflection). The authors claim that UALM achieves state-of-the-art or competitive performance across all three domains in a unified manner.

**Strengths:**

1.  **Ambitious and Visionary Goal:** The paper sets out with a highly ambitious and forward-looking vision: to create a single, unified model for audio intelligence that integrates understanding, generation, and reasoning. This conceptual goal is well-motivated by parallels with human cognition and represents a significant step towards more general and capable AI systems.

2.  **Impressive System-Building and Engineering:** The sheer scale and complexity of the engineering effort are commendable. The authors successfully tackle numerous technical challenges, from scaling up a language model-based audio generator to rival diffusion models, to carefully balancing a multi-task, multi-modal training regime. The methodical approach, breaking down the grand vision into manageable technical problems and providing effective solutions for each, is a major strength.

3.  **Pioneering Exploration of Generative Reasoning:** The introduction of multimodal reasoning for audio *generation* is the most novel and exciting contribution of this work. The concept of "Self-Reflection"—a cycle of generating, understanding, critiquing, and refining—is a powerful paradigm that mimics human creativity and problem-solving. This pushes the boundary beyond simple input-output generation towards more intelligent, iterative systems.

**Weaknesses:**

Despite the impressive scope and results, the paper suffers from several significant weaknesses that temper its contributions and raise questions about the validity of some of its core claims.

1.  **Misleading Performance Claims Due to External Enhancement Module:** The paper's headline claim of achieving state-of-the-art generation quality is confounded by its reliance on a powerful, external "Enhancement VAE." This module, which upsamples the model's 16kHz mono output to 48kHz stereo, is a separate, large post-processing network. The ablation study in Table 8 reveals that this VAE is responsible for a massive improvement in objective scores (e.g., FD). This raises two critical issues:
    *   **Lack of Fairness:** The performance of the core `UALM` is conflated with the performance of this post-processor. Comparisons against other SOTA models are unfair unless those models are also afforded the same enhancement module. The true generative capability of the unified model itself remains unclear.
    *   **Contradiction of "Unified" Claim:** The necessity of a separate, large enhancement module contradicts the central premise of achieving all capabilities "in a single model." A truly unified model should be able to generate high-fidelity audio end-to-end.

2.  **Lack of Evident Synergy in the "Unified" Model:** The paper’s foundational argument is that unifying understanding, generation, and reasoning will create a synergistic system where abilities mutually reinforce each other. However, the paper provides little to no evidence that this synergy occurs during the unified pre-training of the main `UALM` model. The training appears to be a form of multi-task learning, demonstrating that a single model *can* perform these tasks concurrently, but not that it performs them *better* because of the unification. The most compelling synergy (Self-Reflection) only emerges after extensive, specialized post-training in `UALM-R1`, which calls into question the actual benefits of the unified pre-training stage itself.

3.  **Incomplete Literature Review:** The paper's literature survey for audio generation is not up-to-date. It omits several recent and highly influential text-to-audio generation models, such as **Make-An-Audio** and **AudioLCM**. These models represent the cutting edge of the field. To robustly claim that a language model-based approach can be "comparable to state-of-the-art diffusion-based models," it is essential to benchmark against these latest and strongest competitors. The current comparison set, while respectable, is incomplete.

4.  **Heavy Reliance on a Non-Transparent Intermediate Representation:** The advanced reasoning capabilities of `UALM-R1` are heavily dependent on a manually designed intermediate representation, the "Rich Caption." This structured format simplifies the reasoning task into what can be seen as a sophisticated form of "template filling." This raises questions about the generality of the learned reasoning. It is unclear how the model would perform on complex tasks that cannot be easily decomposed into the predefined "Keywords, Layout, Description" structure. This suggests the model may be learning a specific workflow rather than a general, flexible reasoning ability.

**Questions:**

1.  Given the significant performance boost from the Enhancement VAE, would it be more transparent to present the primary results of `UALM-Base` (without enhancement) and compare those directly to other models? How does the core unified model's raw output quality stand against SOTA baselines before any post-processing?
2.  The paper argues for the "synergy" of unification. Can you provide any analysis or evidence showing that the unified pre-training of `UALM` leads to improvements in one capability (e.g., generation) as a direct result of being trained on another (e.g., understanding), compared to training them separately on the same data?
3.  The "Self-Reflection" mechanism is a highlight of the paper. However, it operates on a macro, "post-hoc" level. Have you considered or explored more fine-grained, internal self-correction mechanisms that could operate during the token-by-token generation process itself?
4.  The success of the reasoning abilities seems tied to the structure of the "Rich Caption." How robust is this approach to tasks that do not fit neatly into this predefined structure? Does the model exhibit more general, free-form planning capabilities beyond this format?

---

> ### Author Response · Authors · 2025-11-21
> **Reply to the reviewer**
>
> - W1 & Q1 (External Enhancement Module ...)
>   - We would like to clarify that the Enhancement VAE only improves the poor acoustic quality of X-Codec decoder. It does not generate any novel sound; all the semantic contents and novel sounds are generated via UALM core component. In other words, our Enhancement VAE functions as an improved X-Codec decoder that outputs 48kHz stereo audio with less artifacts. Note that two stage waveform decoding strategy is a common practice LM works generating auditory signals (e.g., YuE, Kimi-Audio), where the main module generates semantic audio contents and a second-stage module (e.g. super-sampler) focuses on acoustic quality.
>   - FD (OpenL3) measures audio quality at 48kHz stereo, and it really punishes low sampling rate audio. If we conduct 16kHz downsampling of our UALM-Gen outputs (last row), the FD is 201.46 for SongDescriber and 250.48 for AudioCaps, which are similar to the "+DPO" model (see Table 8, Appendix C.2). We also added evaluation results of ground truth after X-Codec reconstruction in Table 8, and FD is very high. These prove that FD is sensitive to sampling rates, and the diffusion baselines with native 44/48kHz outputs already match the FD's evaluation setup. We also note that baselines do not benefit from this Enhancement VAE because it is designed for X-Codec artifact restoration only and the baselines can already output HiFi audio. Running our Enhancement VAE for baselines will degrade their outputs.
>   - In addition, our Enhancement VAE does not improve much on other 16kHz evaluation metrics, and our model already matches or outperforms SOTA on many of them. In summary, our Enhancement VAE does not hack the numbers, but just fixes the X-Codec output artifacts. We acknowledge that the current writing could lead to misleading, and have updated Table 8 in Appendix C.2 for better clarification. We include sampling rates of each baseline as well as each objective metric for direct comparison, and it also includes results without Enhancement VAE.
>
> - W2 & Q2 (Lack of Evident Synergy...)
>   - We appreciate this comment, and we would like to clarify that our claim is: *Concurrent generation and understanding capabilities within one single model is necessary to unlock high-level intelligence in audio*, and we do not intend to show in this work that *joint training on understanding and generation tasks could directly facilitate each other under existing evaluation scheme*, which we aim to achieve in our future work. Per your suggestion, we have replaced the word **synergy** with **"a joint modeling scheme"** to avoid ambiguity.
>   - We would like to claim that the **joint modeling scheme** enables critical capabilities that were impossible in single-task models. For instance, the "Self-Reflection" mechanism (Section 2.4) allows the model to generate, listen, think, and then revise. We aim to show that with our unified pre-training, one can achieve any interleaved audio-text format as long as such post-training data are curated.
>
> - W3 & Q3 (Incomplete Literature Review)
>   - Thanks for the suggestion. We will include Make-an-Audio, AudioLCM, and several additional baselines in Table 1. We are working on setting up the proper environments of these baselines and will report results and update our manuscript by the end of the discussion period.
>
> - W4 & Q4 (Reliance on a Non-Transparent Intermediate Representation)
>   - This is a very interesting direction. We think it also fits the general interleaved audio-text templates despite in a very specific format. Therefore, we might be able to achieve this via proper data curation tailored for this setup. We will investigate this direction in our future research.

---

> > ### Comment · Reviewer_Gqug · 2025-11-23
> > **Response to Authors**
> >
> > Thank you for the authors' response. It fully addresses my previous concerns, and I will maintain my positive score.

---

> ### Author Response · Authors · 2025-11-24
> **Response (Continued)**
>
> Dear Reviewer:
>     To address "W3", we have included additional baselines (Make-an-Audio and AudioLCM) in Table 8 for comparison. Note that our proposed UALM is still competitive given these baselines.
>     Thank you again for recognizing our work.
>
> best,
> UALM authors

---

### Official Review · Reviewer_BqjX · 2025-11-10

**Soundness:** 3
**Presentation:** 3
**Contribution:** 3
**Rating:** 6
**Confidence:** 5

**Summary:**

The paper proposes a audio language model that unifies audio understanding, text-to-audio generation, and text-only reasoning within a single framework. The model jointly learns to perceive, generate, and reason across audio and text modalities by extending a text LLM with modal alignment and post-training. The paper also introduce a caption-rich multimodal dataset that links audio with diverse textual descriptions, which enables downstream tasks, such as controllable audio generation. Experiments demonstrate competitive results on benchmarks for audio understanding, text-to-audio generation, and text-based reasoning. The paper is well written.

**Strengths:**

*Unified multimodal design and empirical performance*. The paper presented a unified framework for audio understanding, text-to-audio generation, and reasoning. The proposed model achieves competitive or comparable results across diverse tasks, showing that the multimodal post training is effective.

*Data contribution*. The introduction of a caption-rich dataset with detailed text descriptions for audio clips is an important resource. It supports both controllable generation and fine-grained understanding, addressing a bottleneck in current audio-language research.

**Weaknesses:**

*Description of data curation*. As described in the paper, the desirable performance appears to rely heavily on the richness of the captions; however, the process of dataset curation remains unclear. A more detailed explanation beyond Sections 2.4.2 and 3.1 would strengthen the paper, particularly regarding data filtering, caption generation, and quality control.

*Preference design in stage 1*. I am somewhat concerned about the use of the CLAP model for DPO training, as it may not accurately capture human preferences in text–audio alignment. The authors could consider or at least discuss preference-aware alternatives such as Human-CLAP [1].

*Lack of ablation*. It is also unclear how much the model benefits from each post-training stage (e.g., instruction tuning, alignment tuning, DPO). Providing ablation studies or quantitative comparisons would help clarify their relative contributions.

Other minor issue:
*Comparison with speech language model* It is recommended that the authors discuss speech language models (SLMs) and clarify how their proposed framework differs conceptually and technically from SLMs.

**Questions:**

The authors are encouraged to address or answer the question raise in Weaknesses.

---

> ### Author Response · Authors · 2025-11-21
> **Reply to reviewer**
>
> - Q1 (Description of data curation).
>   - We have provided a more detailed description of the data source, filtering, labeling, and quality control in Appendix E in the updated manuscript.
>
> - Q2 (Preference design in stage 1).
>   - We thank the reviewer for this constructive suggestion. We agree that employing a human preference-aware metric, such as Human-CLAP [1], could further enhance alignment with human perception. We have updated our Related Work (Section 4) to explicitly acknowledge this potential improvement. However, due to the lack of publicly available checkpoints for Human-CLAP, we were unable to incorporate it into the current study.
>   - We would also like to point out that using standard CLAP models for DPO pair selection is a proven strategy. Recent state-of-the-art text-to-audio works like Tango2 [2] and TangoFlux [3] adopt this strategy and show consistent improvements in both subjective and objective metrics.
>   - [1] Taisei Takano et al, Human-CLAP: Human-perception-based contrastive language–audio pretraining, in APSIPA 2025.
>   - [2] Navonil Majumder, et al. Tango 2: Aligning diffusion-based text-to-audio generations through direct preference optimization.
>   - [3] Chia-Yu Hung, et al, TangoFlux: Super fast and faithful text to audio generation with flow matching and clap-ranked preference optimization.
>
> - Q3 (Lack of ablation).
>   - We thank the reviewer for mentioning this important point. We kindly remind the reviewer that we have conducted ablation study in Table 8 in the Appendix C.2. Results indicate DPO significantly improves KL, CLAP, IS, and AES. FD is sensitive to sampling rate and is upper bounded by X-Codec's decoding quality, so DPO does not improve much on that.
>   - In multimodal post-training, each step of Fig. 3 aims to add a new capability to the model. Since there is no existing evaluation protocol for these complex tasks, we manually inspect the generated reasoning traces and audio samples after each step to make sure our model behaves as expected. We conduct human evaluation in Table 4 and show that the multimodal post-trained model achieves better results on all three capabilities.
>
> - Q4 (Other minor issue)
>   - Thanks for the great suggestion. We have added additional discussion in "Related Work" (Section 4) regarding speech language models.

---

### Author Response · Authors · 2025-11-21
**Thanks to AC and all reviewers**

We thank AC for organizing this discussion. We thank all reviewers for their valuable time in reviewing our paper and their constructive comments. We highlight our revisions in **blue** font in the updated pdf.

---

### Meta-Review · Area_Chair_YVi8 · 2026-01-08

**Summary:**

The paper proposes UALM, a unified audio language model that integrates audio understanding, text-to-audio generation, and multimodal reasoning within a single framework. It also introduces UALM-R1 for audio generative reasoning via enrichment, dialogue, and self-reflection. Reviewers largely agree that the work is ambitious, well-engineered, and empirically strong, demonstrating competitive or SoTA results across multiple benchmarks for audio understanding and generation. The most debated aspects concern the evaluation and interpretation of multimodal reasoning, dataset transparency, reliance on an enhancement VAE for audio quality, and the lack of standardized benchmarks for audio reasoning. Overall, reviewer sentiment is strongly positive, with a clear majority recommending acceptance. AC agrees with the sentiment and feels the reviewer scores would have improved after the rebuttal.

**Reviewer Concerns:**

Concerns addressed by the rebuttal:
1/ Fairness of comparisons and missing parameter counts for baselines (tables updated).
2/ Dataset construction, filtering, and reproducibility (detailed in Appendix E).
3/ Role and impact of the Enhancement VAE (clarified as an improved codec decoder; results without enhancement added).
4/ Lack of ablations for training stages and alignment warm-up (existing and new ablations clarified).
5/ Incomplete literature coverage (additional baselines such as Make-An-Audio and AudioLCM added).
6/ Overfitting to evaluation metrics (clarified use of distinct models for DPO selection vs. evaluation).

Remaining limitations:
1/ Evaluation of multimodal audio reasoning (UALM-R1) relies on small-scale human studies due to lack of established benchmarks.
2/ Reasoning capabilities depend on structured rich caption representations, with unclear generality beyond this format.

**Reviewer Scores:**

Based on the rebuttal and discussion, I expect the following score changes:
Reviewer BqjX: 6 to 7
Reviewer Gqug: 6 to 7
Reviewer jHYo: 6 to 7
Reviewer jfVq: 8 to 8
Reviewer tZVa: 8 to 8
Reviewer 9tTC: 2 to 4

Overall score distribution strongly favors acceptance.

---

### Decision · Program_Chairs · 2026-01-26

Accept (Oral)